# Slow Breathing Reduces Biomarkers of Stress in Response to a Virtual Reality Active Shooter Training Drill

**DOI:** 10.3390/healthcare11162351

**Published:** 2023-08-21

**Authors:** Courtney C. Dillard, Hunter Martaindale, Stacy D. Hunter, Matthew J. McAllister

**Affiliations:** 1Metabolic & Applied Physiology Lab, Texas State University, San Marcos, TX 78666, USA; 2ALERRT Center, Texas State University, San Marcos, TX 78666, USA

**Keywords:** slow breathing, military, law enforcement, epinephrine, catecholamines, autonomic nervous system, stress, virtual reality

## Abstract

Tactical occupations regularly encounter life-threatening situations while on duty. Although these occupations are often trained to utilize slow breathing (SB) during intense stress, there is no evidence supporting the effects on markers of stress in response to a virtual reality active shooter training drill (VR-ASD). The purpose of the study was to determine the impact of acute SB on biomarkers of stress in response to a VR-ASD. Seventy-nine (n = 79) subjects performed either slow breathing method 1 (SB1), slow breathing method 2 (SB2), or normal breathing (control) for five minutes, both pre- and post-VR-ASD. Saliva samples were analyzed for stress markers, including α-amylase (sAA) and secretory immunoglobulin-A (SIgA). Both methods of SB resulted in significantly lower sAA concentrations at 5 (*p* < 0.001) and 30 min post-VR-ASD (SB1: *p* = 0.008; SB2: *p* < 0.001) compared to the control. In the control condition, the sAA concentrations were significantly elevated 5 min post-VR-ASD (*p* < 0.001) but did not change across time in SB1 or SB2 (*p* > 0.05). Thus, both SB1 and SB2 reduced the sAA response and resulted in lower concentrations post-VR-ASD. This study was pre-registered as a clinical trial (“Impact of Breathing Interventions on Stress Markers”; NCT05825846).

## 1. Introduction

High-stress occupations such as firefighters, military, and law enforcement personnel are regularly exposed to high-risk environments that require individuals to perform under stress [1]. The autonomic nervous system (ANS) plays an intricate role in how the body responds and adapts to physical and psychological stressors. The ANS is divided into two divisions, the sympathetic (SNS) and parasympathetic (PNS) nervous systems, and aids in regulating the cardiorespiratory systems [2,3,4]. Exposure to physiological and psychological stressors activates the hypothalamic pituitary adrenal (HPA) [5] and sympathoadrenal (SA) [6] axes, resulting in increases in blood pressure (BP), heart rate (HR), and respiratory rate (i.e., components of the sympathetic or fight or flight response). While acutely beneficial, chronic exposure to stressors results in cardiovascular strain, oxidative stress, and insulin resistance, which increases the risk for developing clinical disorders such as cardiometabolic and neurodegenerative diseases [6,7,8,9]. In fact, elevated rates of prevalence and risk factors for cardiometabolic disease have been reported in firefighters [10,11] and law enforcement officers [12,13], which may be largely attributed to chronic or intense stress exposure. Consequently, interventions are needed to mitigate the physiological responses to stress for improved occupational performance and to maintain one’s long-term psychological and physiological health. 

Acute participation in high-stress tactical scenarios has been shown to increase blood and salivary markers of stress [14,15,16]. For example, a realistic short-duration (~50 s) active shooter training drill involving professional actors playing the roles of victims and an active shooter resulted in significant increases in markers of HPA and SA activation, including blood epinephrine, norepinephrine, salivary cortisol and α-amylase (sAA), and secretory immunoglobulin-A (SIgA) [14]. Virtual-reality- and video-based high-stress tactical scenarios such as high-speed motorcycle suspect chases and active shooter training scenarios have also been shown to acutely increase markers of stress and inflammation [15,16,17,18]. In fact, a recent study demonstrated that acute exposure to a virtual reality active shooter training drill (VR-ASD) results in the same magnitude of stress response as compared to the same scenario-based training involving professional actors [18]. While virtual reality has been used as an effective tool for stress inoculation or habituation training among law enforcement and military personnel [19,20], interventions that can be used to acutely mitigate the stress responses are needed for improved performance and decision-making and to mitigate the physiological effects.

There is growing evidence that breathing manipulation can help manage the physiological response to stress and cause improved brain function and mood [21,22]. Slow breathing techniques have been traditionally used with law enforcement and military personnel for maintained cognitive performance and stress mitigation [23,24,25]. A slow breathing rate is typically defined as a rate of four to ten breaths per minute, and this usually results in decreases in heart rate and blood pressure when comparing slow breathing versus a normal breathing rate (10–20 breaths per minute) [26,27]. Slow breathing has also been shown to cause increased parasympathetic activity at rest, resulting in a reduced heart rate [28], increased heart rate variability [29,30], increase venous return [31], and decrease the systolic and diastolic blood pressure rates [32]. Moreover, it has also been shown that differences in the inhalation/exhalation ratio can impact markers of sympathetic or parasympathetic activity [33,34,35,36]. For example, the findings by Komori demonstrated increased markers of parasympathetic activity during prolonged exhalation (6:4 exhale to inhale ratio) compared to rapid breathing (1:1 exhale to inhale ratio), demonstrated by changes in high- and low-frequency markers of heart rate variability (HRV) [34]. These findings align with a similar study that demonstrated increased HRV when comparing rapid breathing (1:1 exhale to inhale ratio) to a longer exhalation protocol (2:1 exhale to inhale ratio) [37] However, there are a lack of data regarding the physiological biomarkers of stress. Therefore, the purpose of this study was to determine the impacts of two styles of slow breathing with different inhalation-to-exhalation ratios on the physiological biomarkers of stress (sAA, SIgA, HR) and subjective markers of psychological stress, utilizing the state–trait anxiety inventory (SAI), in response to a VR-ASD that was previously shown to increase these stress markers [16,18]. Based on the findings from previous work [34,37,38,39], we hypothesized that slow breathing would reduce the biomarkers of stress and that this effect would be more pronounced with prolonged exhalation.

## 2. Materials and Methods

### 2.1. Subjects and Experimental Design

This experiment was conducted using a randomized, single-blinded, between-subjects, parallel design to determine the impacts of acute slow breathing on markers of stress in response to a VR-ASD. Fifty-two (n = 52) subjects were randomly assigned to perform either slow breathing 1 (SB1; n = 25) or slow breathing 2 (SB2; n = 27) for five minutes both before and after participating in a VR-ASD. The randomization for the breathing intervention was such that the first two consecutive subjects received a random number (odd = SB1; even = SB2) using a random generator (random.org) and the subsequent two subjects received the opposite treatments. The control (n = 27) consisted of data from a previous study where the subjects performed no modified breathing whatsoever (control group) [16]. The subjects provided electronic informed consent and completed a health history questionnaire prior to participating in the study. The subjects were recruited from a university campus and were aged 18–39 years old, with no known cardiovascular or metabolic diseases; free from any major stressors in the last 30 days such as the birth of a child, abortion, or divorce; with no history of motion sickness or vertigo; and had never been diagnosed with a brain injury or epilepsy. The subjects who qualified to participate self-reported their activity levels (exercise at least 3x/week = 41; no = 11). Each subject arrived 4-h-fasted and was asked to avoid participation in strenuous physical activity 24 h prior to testing. The testing occurred in the afternoon (~12:00–17:30). The procedures utilized were all reviewed and approved by the university’s institutional review board. All subjects who participated in the current study had not been previously involved in past research involving a VR-ASD. 

### 2.2. Experimental Procedures

The procedures mimicked established protocols [14,16,40]. The subjects arrived at the testing site, located at the Advanced Law Enforcement Rapid Response Training (ALERRT) center. Upon arrival, they were asked to rinse their mouth with bottled water and sit and rest in a quiet room for 10 min. After the 10 min rest period, the subjects provided their first saliva sample, HR measurement, and SAI. A saliva sample, HR measurement, and SAI were collected four times: (1) 30 min prior to, (2) 5 min prior to, (3) 5 min post-, and (4) 30 min post-VR-ASD. 

### 2.3. Virtual Reality Study Procedure and Active Shooter Drill

The VR-ASD utilized was based off a previous live-action ASD and involved multiple victims and one gunman [14]. A lab assistant fitted the VR headset and equipped the subjects with a VR-specific training pistol. To ensure the subjects understood how to operate the equipment, they participated in a VR familiarization protocol about 1 min prior to starting the VR-ASD. Upon completion of the familiarization protocol, the subjects were instructed by the researcher that they were acting as an officer on duty responding to an active shooter and their goal was to locate and stop the active shooter by ‘firing’ the training weapon in the VR environment. The VR-ASD was assessed using SSVR software and firearms using HTC VIVE Pro VR equipment (HTC Corp, New Taipei, Taiwan). The VR lab was larger than the virtual environment (~35 × 20 ft). This allowed the subject to physically walk down a ~10-foot-long virtual hallway. The subject continued to walk until arriving at the threshold of the room where the shooter was located and actively firing his handgun at the last victim. If the subject had not fired their weapon yet, the shooter turned toward the subject to elicit a response.

### 2.4. Breathing Interventions

The subjects were randomly assigned to perform either slow breathing 1 (SB1; four second inhalation, two second pause, four second exhalation, two second pause) or slow breathing 2 (SB2; four second inhalation, two second exhalation) for five minutes both before and after participating in the VR-ASD. SB1 resulted in a respiratory rate of five breaths per minute and SB2 resulted in a respiratory rate of ten breaths per minute. Therefore, both SB1 and SB2 were within what is defined as a slow breathing range of four to ten breaths per minute [27]. However, both breathing methods were chosen based on speculation that the differences in the ratio of inhalation to exhalation would affect markers of stress [33,34,41]. Subjects who performed their normal breathing performed no modified breathing (i.e., control group). 

The lead researcher provided each subject with a 5 min instructional video for their assigned condition (i.e., SB1 or SB2). The instructional video provided a visual of the researcher performing the breathing intervention alongside a voiceover with a countdown on the screen for the subjects to keep the correct breathing rhythm. A researcher was in the room observing the subject and was informed to discontinue the breathing intervention if the subject was unable to follow along with the breathing intervention or if the subjects reported any other issues (i.e., vertigo, fainting, lightheadedness). The subjects were informed that they could voluntarily withdraw from the study at any point. The subjects performed the randomly assigned breathing intervention immediately before the VR-ASD and immediately post-VR-ASD. 

### 2.5. State-Anxiety Inventory Assessment and Heart Rate

Measures of SAI and HR were collected concurrent with each saliva sample (30 min pre-, 5 min pre-, 5 min post-, and 30 min post-VR-ASD). The SAI had been previously established as a reliable and valid scale to assess subjective stress [42,43]. The SAI included six short statements such as “I feel calm” and “I am tense”, scored on a scale of 1–4. A composite score for each scale was utilized for the analysis. The HR was assessed via a blood pressure monitor (American Diagnostic Corporation, Hauppauge, NY, USA). The subjects were instructed to sit in a chair and not cross their legs or talk while getting their HR measurements. 

### 2.6. Saliva Collection and Analysis

The subjects collected their own saliva using the passive drool method using a saliva collection tube and aid (Salimetrics, PA, USA). Each sample (~1.5 mL) was gathered by allowing saliva to pool in the mouth, tilting the head forward, and gently guiding the saliva into the tube. The saliva samples were collected 30 min pre-, 5 min pre-, 5 min post-, and 30 min post-VR-ASD. Approximately 10 min prior to each sample (besides the 5 min post-VR-ASD), the subjects were asked to rinse their mouth with water. The saliva was immediately stored in a compact ultra-low portable freezer (Cole-Parmer, IL, USA) at −80 °C until the analysis. The saliva was taken back to the Metabolic and Applied Physiology Laboratory at Texas State University once all subjects had completed the study for the day. For the data analysis, the samples were thawed, centrifuged at 4 °C for 15 min, and transported overnight on dry ice to a laboratory, then subsequently analyzed in duplicate for concentrations of sAA and SIgA using commercially available kits (Salimetrics, PA, USA). The intra-assay coefficient of variation (CV) range was 5.4–5.6% and the inter-assay CV range was 4.7–8.9% for sAA and SIgA.

### 2.7. Statistical Analysis

All statistical analysis procedures were conducted using SAS v 9.4 (Cary, NC, USA). Separate 3 × 4 (treatment × timepoint) factorial ANOVAs were used to compare the changes across time and between conditions for sAA and SIgA. Data were not available for HR and SAI values for the control condition; therefore, changes across time and between treatments for the SAI and HR were analyzed via a 2 × 4 (treatment × timepoint) factorial ANOVA. In the instance of a main effect or interaction (*p* < 0.05), Fisher’s LSD test was conducted. The effect sizes were calculated and reported as the partial eta (ηp^2^). The a priori power analyses utilized the smallest effect from the control group (partial eta square = 0.14) to estimate the sample size needs for the present study. G*Power 3.1.9.7 estimated that a total sample size of 63 was needed to achieve a power of 0.80 to detect an effect. As such, we attempted to match the sample size in the control group (n = 29) in order to oversample (thereby increasing the power to detect effects).

## 3. Results

A total of 52 subjects (n = 52; male = 22; female = 30; body mass: 72.8 ± 17.3 kg; height: 168.1 ± 11.1 cm; age: 22 ± 4 y; baseline systolic blood pressure: 132 ± 16; baseline diastolic blood pressure 78 ± 14) completed the experimental testing for SB1 (n = 25) and SB2 (n = 27). The subject demographics by treatment group are demonstrated in Table 1, and subjects that completed the experimental testing for SB1 and SB2 are shown in Figure 1. A total of 27 subjects were used for the control condition (n = 27; males = 17; females = 12) [16]. All data are shown as means ± standard deviations unless otherwise noted.

### 3.1. Saliva Data

Regarding sAA, a significant treatment × time interaction (F = 5.07, *p* < 0.001, ηp^2^ = 0.12) was found. The post-hoc analysis demonstrated significantly higher sAA concentrations immediately post-VR-ASD in the control condition compared to SB1 and SB2 (*p* < 0.001). In addition, the sAA concentrations at 30 min post-VR-ASD were significantly higher in the control condition compared to SB2 (*p* < 0.001) and SB1 (*p* = 0.008). With respect to the changes across time, the control condition demonstrated a significant increase in sAA immediately post-VR-ASD (*p* < 0.001) compared to all other timepoints, whereas both SB1 and SB2 demonstrated no changes across time for any timepoint (*p* > 0.05). The mean SAA concentrations are shown in Figure 2.

### 3.2. SIgA

With respect to the mean SIgA concentrations, no treatment × time interaction was noted (*p* > 0.05). However, there were main effects for the treatment (F = 105.2, *p* < 0.001, ηp^2^ = 0.40) and time (F = 15.05, *p* < 0.001, ηp^2^ = 0.12). The post-hoc analysis demonstrated significantly higher SIgA concentrations in the control condition compared to both SB1 and SB2 (*p* < 0.001). In addition, significant increases in SIgA concentrations were demonstrated immediately post-VR-ASD compared to all other timepoints (*p* < 0.001). The mean SIgA concentrations are shown in Figure 3.

### 3.3. Heart Rate

With respect to the mean HR, no treatment × time interaction was noted (*p* > 0.05). However, there were main effects for the treatment (F = 24.9, *p* <0.001, ηp^2^ = 0.14) and time (F = 3.99, *p* = 0.009, ηp^2^ = 0.07). The post-hoc analysis demonstrated a significantly higher mean HR in SB2 compared to SB1 (*p* < 0.001). In addition, the HR at 30 min pre-VR-ASD was significantly higher than all other timepoints. The mean HR data are shown in Figure 4.

### 3.4. State Anxiety Inventory

With respect to the mean SAI data, no treatment × time interaction was found (*p* > 0.05). There was no main effect for the treatment (*p* > 0.05). However, there was a main effect for time (F = 4.45, *p* = 0.005, ηp^2^ = 0.08). The SAI values were significantly lower at 30 min post-VR-ASD compared to 5 min post-VR-ASD (*p* = 0.002) and 30 min pre-VR-ASD (*p* = 0.003). The mean SAI data are shown in Figure 5.

## 4. Discussion

The main findings of the study demonstrate that slow deep breathing (both SB1 and SB2) prevented significant increases in sAA concentrations in response to stress and resulted in lower sAA concentrations post-VR-ASD. While the SIgA concentrations were lower in the SB1 and SB2 conditions, these differences were likely not due to the breathing intervention, since there was no significant interaction. The present study was the first to examine the effects of slow deep breathing on stress biomarkers in response to a simulated virtual-reality-based tactical high-stress scenario. Future studies should examine the long-term effects of slow deep breathing in relation to stress and cardiovascular disease risk factors.

It should be noted that a key difference between SB1 and SB2 is that the exhalation phase was shortened for SB2 relative to SB1, such that it included a 4:2 inhale to exhale ratio and SB1 included a balanced 4:4 inhale to exhale ratio. Our team hypothesized that SB2 would demonstrate elevated markers of stress due to reduced CO_2_ clearance in the periphery [44] and a reduced ratio of exhalation relative to inhalation [34]. However, the current results do not support that hypothesis. In fact, SB1 and SB2 were equally as effective at reducing the sAA concentrations. This was likely due to the fact that both SB1 and SB2 were voluntarily controlled by neural inputs [45] and were within what is defined as a slow breathing range of 4–10 breaths per minute [27]. Similarly, a previous study that suggested that the rate of slow breathing is more influential on autonomic responses compared to the ratio of inspiration to expiration [46]. During inspiration, the activation of the stretch receptors of the lungs (i.e., the Hering–Breuer reflex) [47] increases the inhibitory neural impulses that contribute to the control of the autonomic nervous system (i.e., breathing rate, heart rate) [48] through the synchronization of neural elements (i.e., hypothalamus, brainstem), leading to parasympathetic dominance [49]. Therefore, this is one mechanism that could potentially explain the current findings.

Acute stress exposure increases the concentrations of sAA, which reflects increased activity of the SNS [50,51,52]. Although activation of the SNS is a defensive response to increase an individual’s chance of survival and to maintain homeostasis [53], significant increases in sAA are associated with acute increases in anxiety and decreased cognitive performance. The observed increase in sAA in response to the VR-ASD (in the control condition) was likely due to the activation of the SNS in response to the VR-ASD [34,54,55,56,57]. Previous research observing sAA and cardiac vagal (parasympathetic) activity levels during high- and low-stress shooting scenarios resulted in significant increases in sAA during both scenarios [58]. The findings from the present study demonstrate that slow breathing attenuates the sAA response to acute stress. These findings are in line with previous studies that demonstrated that a slow breathing rate (8 breaths per minute) compared to a more rapid breathing rate (12–18 breaths per minute) resulted in an increase in heart rate variability, reflecting increased parasympathetic activity, decreased sympathetic activity, and decreased blood pressure [29,30,59]. On that account, the reductions in sAA observed in SB1 (5 breaths per minute) and SB2 (10 breaths per minute) in response to the VR-ASD can likely be attributed to reductions in sympathetic activity for both breathing interventions.

The present study demonstrated increases in SIgA concentrations resulting from exposure to the VR-ASD, which is in agreement with past work [16,18]. SIgA is a dominant antibody in saliva for the defensive immune function [53]. Chronic stress is known to reduce the concentrations of SIgA [60,61], while acute stressors (i.e., mental arithmetic, computer games, memory recall) increase the concentrations of SIgA [62,63,64,65]. The relationship between mental stress and the immune function is not fully understood [66]. As expected, there were significant increases in SIgA levels in response to the VR-ASD due to an acute psychological stressor, which has been supported previously [14,16]. The findings from the present study are in line with the findings from a past study that showed no change in SIgA concentrations in response to acute stress when performing slow breathing [67]. Past work has suggested that SIgA is mediated by sympathetic activity [68]; thus, the lack of an effect from slow breathing suggests that slow deep breathing impacts the sympathetic or parasympathetic activity but not the immune activity. Therefore, future research should determine if longitudinal participation in slow breathing reduces the SIgA levels to further expand on the effects slow breathing has on the immune function.

Regarding the changes in HR and SAI between conditions and across time, the results demonstrate a significant elevation of the HR 30 min prior to the VR-ASD, as well as a reduction in the SAI 30 min post-VR-ASD. Both of these suggest decreases in HR and SAI across time, which could potentially be related to stress or anxiety associated with anticipation of participation in the experimental procedures. During states of anxiety or psychological stress, a reduction in vagal (parasympathetic) activity can be reflected by an increase in heart rate as well as an increased respiratory rate (i.e., hyperventilation) [69,70,71]. A previous study that required subjects to make judgements or decisions under the threat of an electric shock if too many errors were made resulted in significant increases in HR and SAI from baseline [72]. Thus, it is possible that the VR-ASD was not stressful enough to elicit increases in SAI and HR. However, the lack of an increase in SAI or HR in response to the VR-ASD was likely due to the timeframe of measurements. The HR and SAI values were not collected immediately post-VR-ASD but were collected after completing the breathing interventions. Thus, it is not surprising that we did not observe an increase in HR or SAI in response to stress. However, while the present study is limited by not having HR or SAI data for comparison to the control condition, we hypothesized that due to the differences in the inhalation/exhalation ratios between SB1 and SB2, there would be measurable differences between these conditions with respect to the HR and SAI [33,34]. Since there was no difference in HR and SAI between both SB1 and SB2, this is likely explained by the fact that both of these interventions were within the defined range of slow breathing [27,35] and the reduced exhalation phase in SB2 had no effect.

There were a few limitations of the current study. First, the study was limited by not having HR and SAI data for comparison to the control condition. In addition, the salivary stress, HR, and anxiety measures were not collected both before and after each breathing intervention. Therefore, it was not possible to determine the magnitude of the stress reduction from each 5 min breathing intervention or the actual response from the VR-ASD alone. However, past work from our lab has shown the salivary stress response to the VR-ASD when no breathing intervention was performed [16]. In addition, both the men and women that were enrolled in the present study and these subjects were not trained law enforcement officers. Individuals that are regularly exposed to occupational high-stress scenarios (such as active shooter training simulations) may respond differently to acute stress. Thus, both of these factors (sex, training status) could impact the response to the VR-ASD. Our screening questionnaire only inquired about the subject’s gender; therefore, a secondary analysis on the impact of biological sex was not possible. Finally, while the age range was consistent, demographic data for height and weight were not available for the control group, and this should be viewed as a limitation.

## 5. Conclusions

In conclusion, the VR-ASD resulted in significant increases in the physiological stress markers sAA and SIgA. Both slow breathing interventions (SB1 and SB2) were effective at preventing increases in the salivary stress marker (sAA) in individuals post-VR-ASD exposure. In addition, the sAA concentrations were significantly lower post-VR-ASD. Future research should aim to identify the impacts that slow breathing interventions have on performance and decision-making during a high-stress scenario and the plausibility of incorporating slow breathing techniques during a high-stress scenario versus performing a breathing intervention. Moreover, future work should consider adding additional indicators of ANS function such as skin conductance or heart rate variability.

## Figures and Tables

**Figure 1 healthcare-11-02351-f001:**
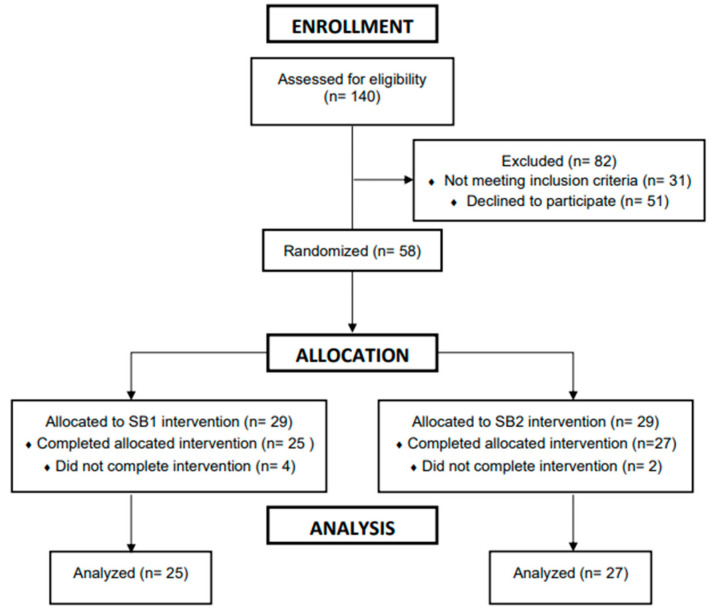
Consort diagram.

**Figure 2 healthcare-11-02351-f002:**
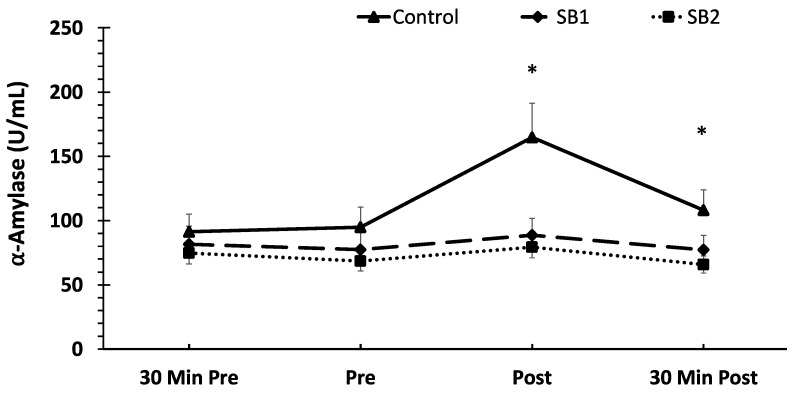
Mean salivary α-amylase (sAA) concentrations across time and between treatments. Data are shown as means ± SE. * Indicates significantly greater sAA concentrations immediately post- and 30 min post-VR-ASD in the control condition compared to SB1 and SB2; 30 Min Pre = 30 min pre-VR-ASD, Pre = immediately pre-breathing intervention; Post = immediately post-breathing intervention; 30 Min Post = 30 min post-VR-ASD.

**Figure 3 healthcare-11-02351-f003:**
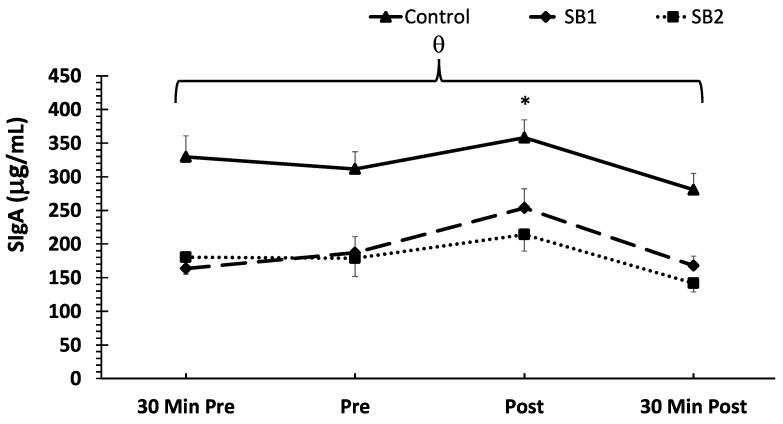
Mean SIgA concentrations across time and between treatments. Note: θ indicates higher SIgA concentrations overall in the control compared to the SB1 and SB2 groups (*p* < 0.001). Data are shown as means ± SE. * Indicates a significantly higher SIgA concentration compared to all other timepoints; 30 Min Pre = 30 min pre-VR-ASD; Pre = immediately pre-breathing intervention; Post = immediately post-breathing intervention; 30 Min Post = 30 min post-VR-ASD.

**Figure 4 healthcare-11-02351-f004:**
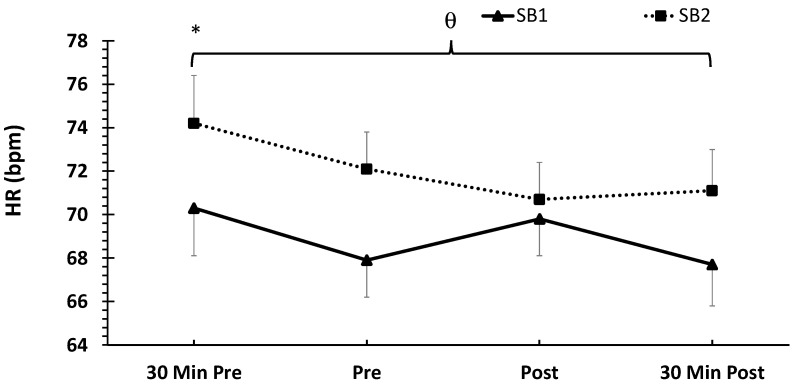
Mean heart rate (HR) responses across time and between treatments expressed in beats per minute (bpm). Data are shown as means ± SE. Note: HR data are not available for the control condition; θ indicates significantly higher mean HR in the SB1 group compared to SB2. * Indicates significantly higher mean HR 30 min pre-VR-ASD compared to all other timepoints; 30 Min Pre = 30 min pre-VR-ASD; Pre = immediately pre-breathing intervention; Post = immediately post-breathing intervention; 30 Min Post = 30 min post-VR-ASD.

**Figure 5 healthcare-11-02351-f005:**
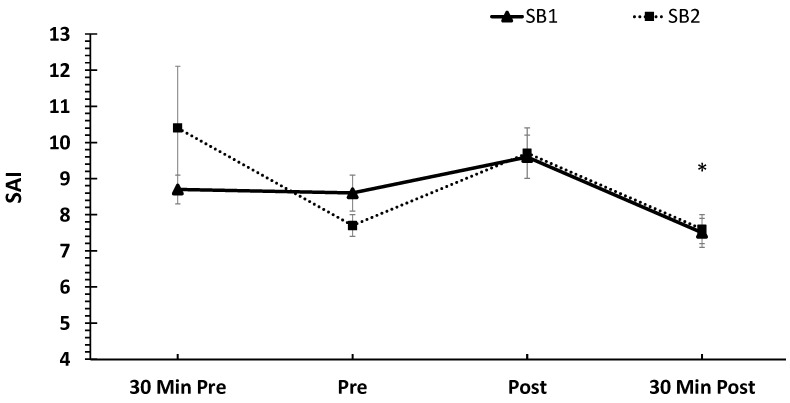
Mean state anxiety inventory (SAI) values across time and between treatments. Data are shown as means ± SE. Note: SAI data are not available for the control condition. * Indicates significantly lower SAI values at 30 Min post-VR-ASD compared to immediately post (*p* = 0.002) and 30 min pre-VR-ASD (*p* = 0.0003); 30 Min Pre = 30 min pre-VR-ASD; Pre = immediately pre-breathing intervention; Post = immediately post-breathing intervention; 30 Min Post = 30 min post-VR-ASD.

**Table 1 healthcare-11-02351-t001:** Subjects’ descriptive characteristics.

	Height (cm)	Mass (kg)	BMI (kg/m^2^)
Slow breathing 1	168.8 ± 9.5	74.8 ± 16.0	26.2 ± 5.4
Slow breathing 2	167.8 ± 12.3	70.9 ± 18.6	24.7 ± 3.6

## Data Availability

The data will be made available upon reasonable request.

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
