# Peer review of "Slow Breathing Reduces Biomarkers of Stress in Response to a Virtual Reality Active Shooter Training Drill"

_healthcare, 2023, doi:10.3390/healthcare11162351_

Round 1

Reviewer 1 Report

The aim of this research was to investigate how acute slow breathing (SB) techniques, specifically SB1 and SB2, affect stress biomarkers in response to a virtual reality-based acute stressor (VR-ASD). A total of 81 participants took part in the study. They were assigned to perform one of three breathing methods: SB1, SB2, or normal breathing (NB) for five minutes before and after the VR-ASD. The researchers analyzed saliva samples to measure stress markers, specifically α-amylase (sAA) and secretory immunoglobulin-A (SIgA).

The results showed that both SB1 and SB2 led to significantly lower sAA concentrations at 5 minutes and 30 minutes after the VR-ASD. In contrast, participants who underwent NB experienced a significant increase in sAA levels at 5 minutes post VR-ASD but showed no significant changes in sAA levels across time during SB1 or SB2. This suggests that both SB1 and SB2 reduced the sAA stress response, resulting in lower concentrations of sAA after the VR-ASD. In summary, the virtual reality-based acute stressor (VR-ASD) led to a notable rise in physiological stress markers, namely sAA and SIgA. However, the implementation of both slow breathing interventions (SB1 and SB2) effectively prevented the escalation of salivary stress marker sAA in individuals after exposure to the VR-ASD. Moreover, the concentrations of sAA were significantly reduced following the VR-ASD when these slow breathing techniques were utilized.

The study is well-conducted and well-written. It was a pleasure reading it. I just have a minor comment:

Comment1# This study's focus on stress levels warrants further exploration of the potential impact of biological sex. While the discussion briefly touched upon this aspect, I recommend conducting a more in-depth analysis or incorporating controls for biological sex in your data to better understand its influence on the observed effects.

Author Response

Please see attached file including our respnoses.

Reviewer 2 Report

In this manuscript, the authors investigated the effect of breathing rate on stress markers sAA and

SIgA in response to virtual -reality active shooter training drill. Overall, while the premise of the

manuscript is intriguing, limitations need to be further addressed for potential interpretation and

publication. Please see individual comments below.

1.       Why choose sAA and SigA to measure out of many stress markers such as epinephrine, norepinephrine? Need more rationality demonstration.

2.       About control group (normal breath): to this reviewer’s understanding, data collected from 27 normal breath control subjects is from previous study (PMC8961715). Control subject characteristics should be included in this manuscript to well demonstrated that data collected from control and slow breath subjects are comparable.

3.       In Figure 3, the baseline of SIgA is very different from the control to SB1/2, this reviewer suggests to not show control group.

4.       From line 173 and line 178, there are 52 slow breathing subjects, and 27 normal breathing subjects, so total are 79 subjects included in the study analysis. But in abstract, authors addressed 81 subjects (line 13) are included in the study.

5.       In figure 1 consort diagram: completed allocated intervention (SB2 group) n=7. Is it a typo?

6.       Figure 3 figure legend, line 211, “q indicates higher SigA concentration….” Where is “q” in the figure?

7.       There is no figure legend for figure 4.

8.       What is “θ” standing for in figure 3 and figure 4?

9.       Please use uniform label in the manuscript: for example: this reviewer assumes “-30 Min Pre” =    -30 = 30 Min Pre, Please just use one label for same meaning.

Author Response

Please see the attached file containing our responses.

Reviewer 3 Report

The purpose of the reviewed manuscript was to analyze how slow breathing reduces stress biomarkers in response to a virtual reality active shooter training drill.

The topic is extremely interesting and it is believed that some points of interest can be suggested to the authors.

- In the general areas, from line 58 to line 66 it would be advisable to integrate some reviews that better describe the relationship between breathing and performance, for example a comprehensive review: https://doi.org/10.3390/sports11050103

- In lines up to 72 for an updated quantitative overview, you can consult a recent meta-analysis on this topic: https://doi.org/10.1080/1750984x.2022.2145573

- The randomisation method is indicated on line 75, it is suggested to insert "randomised" also in the title.

There are no other elements that can create a doubt upon confirmation of publication.

Author Response

(The authors gave the same response as above.)

Reviewer 4 Report

The paper is well written and thorough in describing the methods. The results are clearly presented.

Some issues: The two salivary biomarkers used have been critically reviewed and are often criticized. This should be mentioned in the limitations section.

Another limitation might be the lack of ANS measures (Skin Conductance, HRV). 

Also, the brief nature of the respiratory training should be mentioned. Most findings using slow breathing assess the effects over a training period. 

The literature review does not include the area of heart rate variability biofeedback, which uses similar slow breathing methods for a variety of stress-related conditions. The lack of HR results found in this study is in contrast to many studies in this area. 

Author Response

(The authors gave the same response as above.)

Reviewer 5 Report

General comments

The study is simple but presents some contributions to current knowledge. The authors did not present the sample size estimate for the groups. This is imperative. Present a didactic figure about the experimental design, explaining the interventions and the variables collected in it. The rationale is partially explored; it is subject to improvement. The choice of sample is questionable. It does not match the rationale of the introduction. This needs to be properly discussed. I recommend that all queries be improved in the manuscript, not just answered. The inclusion of the control group is also questionable. The authors probably leveraged results from their previous studies (14, 16, and 35). Be careful with this approach, and present all the participants’ data in the control group.

 Specific comments

Introduction

--The introduction presents simplistic elements of physiology, but overall, the rationale has been developed. However, in the last paragraph, I recommend inserting the hypothesis and supporting it with previous studies. Not enough, in the lines, the authors describe "in response to a stressful stimulus". Describe the stimulus (i.e VR-ASD).

 Methods

--Line 75 – Were the researchers also blinded?

--Line 78 - Only here it was described that two slow breathing protocols would be used. This rationale was not presented correctly in the introduction.

--Line 75-95 - The authors broke the rationale of the study with this paragraph. The relationship between firefighters, military, and law enforcement personnel and stress was made clear in the introduction. Why was this population not recruited for the study? The inclusion criteria are not compatible with the ideas presented in the introduction. I understand that the stimulus (i.e. VR-ASD) is the stressor, but people regularly exposed to these situations may act differently from the people included in this study.

Additionally, why did you select different individuals for the SB1 and SB2 tests? Was the pre-test stress level assessed?

--Line 97-103 - What about pre-test care? Citing previous studies is insufficient. This paragraph needs to be drastically improved.

 Results

--The legends of the figure require improvements. You need to explain again the acronyms. The figures should be self-explanatory. There is no legend in Figure 4. Why the control group was not inserted/compared for HR and SAI?

 --“NORMAL” should be CONTROL, or something similar. “Normal” is relative.

 Discussion

--Line 299-300 – Why?

--Line 309-324 – This discussion should appear before, not here.

--Provide suggestions for future studies.

Author Response

(The authors gave the same response as above.)

Round 2

Reviewer 2 Report

This revision sufficiently addressed this reviewer’s concern.

Reviewer 5 Report

The authors answered all my questions.